# Process of Glucose Increases Rather Than Constant High Glucose Was the Main Cause of Abnormal Glucose Induced Glomerulus Epithelial Cells Inflammatory Response

**DOI:** 10.3390/ijms24010600

**Published:** 2022-12-29

**Authors:** Jiancheng Qi, Weiyu Liu, Linli Gan, Hongrui Guo, Yue Xie, Liping Gou, Dongjie Cai, Jizong Zhang, Junliang Deng, Zhihua Ren, Jing Fang, Zhicai Zuo

**Affiliations:** Key Laboratory of Animal Disease and Human Health of Sichuan Province, College of Veterinary Medicine, Sichuan Agricultural University, Chengdu 611130, China

**Keywords:** diabetic nephropathy (DN), abnormal glycemia, nephritis, apoptosis, pyroptosis

## Abstract

Abnormal glycemia is frequently along with nephritis, whose pathogenesis is unexplicit. Here, we investigated the effects of abnormal glucose on the renal glomerulus epithelial cells by stimulating immortalized bovine renal glomerulus epithelial (MDBK) cells with five different levels of glucose, including low glucose (2.5 mM for 48 h, LG), normal glucose (5 mM for 48 h, NG), high glucose (25 mM for 48 h, HG), increasing glucose (24 h of 2.5 mM glucose followed by 24 h of 25 mM, IG), and reducing glucose (24 h of 25 mM glucose followed by 24 h of 2.5 mM, RG). The results showed that LG and RG treatments had nonsignificant effects (*p* > 0.05) on the viability of MDBK cells. HG treatment decreased the viabilities of cells (*p* < 0.01) without triggering an apparent inflammatory response by activating the *nox4*/ROS/*p53*/*caspase-3*-mediated apoptosis pathway. IG treatment decreased the viabilities of cells significantly (*p* < 0.01) with high levels of pro-inflammatory cytokines IL-1β and IL-18 in the supernatant (*p* < 0.05) by triggering the *txnip/nlrp3/gsdmd*-mediated pyroptosis pathway. These results indicated that the process of glucose increase rather than the constant high glucose was the main cause of abnormal glucose-induced MDBK cell inflammatory death, prompting that the process of glycemia increases might be mainly responsible for the nephritis in diabetic nephropathy, underlining the importance of glycemic control in diabetes patients.

## 1. Introduction

Nowadays, diabetes mellitus (DM) is prevalent worldwide and is the 5th reason for morbidity and mortality [1]. Abnormal glycemia is the main character in DM, including hypoglycemia, hyperglycemia, and glycemia variability, and they increase the levels of the advanced glycation end product (AGE) in the blood [1]. As one of the target organs of hyperglycemia, the kidney often suffers the mesangial apoptosis induced by AGE [2]. The World Health Organization estimated that diabetes patients would reach 592 million by 2035 [3], and over 25–40% of these diabetic patients suffer from nephropathy, named diabetic nephropathy (DN), after 20–35 years [4]. DN, featured by thickened glomerular and tubular basal membranes, accumulation of extracellular matrix, and mesangial cell proliferation, is a severe complication of diabetes [2]. As the first cause of end-stage kidney disease, DN is becoming an increasing health problem and creates a public health challenge along with the worldwide epidemic of diabetes [5].

The pathogenesis of DN is complicated, involving many pathways and mediators, two of which are oxidative stress and inflammatory response [3]. Oxidative stress is an important pathway to the development and progression of DN. Increased reactive oxygen species (ROS), derived from the polyol chain, the AGE, and nicotinamide adenine phosphate dehydrogenase (NADPH) oxidase (Nox), in diabetes, can cause direct damage to podocytes, mesangial cell, and endothelial cells by increasing the levels of proinflammatory cytokines, leading to proteinuria and tubulointerstitial fibrosis [6,7]. Immune and inflammatory responses are thought to play an important role in the pathogenesis of DN, even though that DN has not been considered an inflammatory disease [8]. Moreover, inhibition of the mobilization of inflammatory cells into the kidney showed a protective effect [9]. It is important to note that oxidative stress and inflammation are closely interdependent pathophysiological processes [3,10]. The main pathological mechanisms linking oxidative stress, inflammation, and progression of DN include early renal injury by extra- and intracellular oxygen-derived radicals and its resulting inflammation [11].

In recent years, pyroptosis, a pro-inflammatory event of programmed cell death regulating the innate immune system [12], has been identified as a critical factor in several kidney-related pathological conditions [13], including DN [14]. It is found that gasdermin-D (GSDMD) protein promoted pyroptosis-regulated cell death in high glucose-treated mouse podocytes, and inhibition of GSDMD protein can effectively mitigate high glucose-mediated inflammation and apoptosis by suppressing C-Jun N-terminal kinase (JNK) phosphorylation and mitochondrial ROS generation [15,16]. Although that hyperglycemia was thought to be the inducer of these pathways and mediators contributing to DN, glycemic variability rather than mere hyperglycemia was supposed to play a more pivotal role in conferring additional risks on micro- and macrovascular diabetic complications [17,18]. Glucose instability is found to be independently corrected with cardiovascular-related mortality [19], impairs the recovery of kidney injury [20], and affects vessel healing [21]. However, the underlining mechanisms of glucose fluctuation impairing the recovery of kidney injury are unclear.

We guess that hyperglycemia, hypoglycemia, and glycemia fluctuation might have different effects on the renal glomerulus epithelial cells, which is an important part of the function of the kidney. In this experiment, we treated MDBK cells (immortalized renal glomerulus epithelial cells of adult bovine) with abnormal glucose concentration, confirming that *nox4*/ROS/*p5*3/*caspase-3*-mediated apoptosis and *txnip*/*nlrp3*/*gsdmd*-mediated pyroptosis pathways participated in the hyperglycemia- and increasing glycemia-induced MDBK cells injury, respectively. This work implied that the process of glycemia increases was mainly responsible for the nephritis in diabetic nephropathy, underlining the importance of glycemic control in diabetes patients.

## 2. Results

The viabilities of the MDBK cells treated with different concentrations of glucose are shown in Figure 1B. The HG had the highest influence on cell viability (*p* < 0.001), followed by IG (*p* < 0.001), RG (*p* < 0.05), and LG (*p* < 0.05) compared to the NG treatment. No significant differences were observed between LG and RG (*p* > 0.05) treatments. In contrast, the cell viability of the IG and HG treatments was significantly lower than the RG and LG treatments, indicating that IG and HG treatments had a greater effect on the cell viability of MDBK cells than the (*p* < 0.05) remaining treatments.

The dynamic of the MDBK cells treated with different concentrations of glucose is shown in Figure 1C–E. The LG and RG treatments did not affect the apoptosis and necrosis rates of the MDBK cells (*p* > 0.05). However, HG and IG treatments significantly increased the apoptosis and necrosis rates (*p* < 0.001) in MDBK cells. Notably, HG and IG treatments had different effects on the apoptosis rate (*p* < 0.05), but similar effects on the necrosis rate (*p* > 0.05) were noted. The result of the cell proliferation of the MDBK cells is presented in Figure 1C (Appendix A). There was no significant difference (*p*>0.05) was noted in the AV^−^/PI^+^ quadrant among the 5 treatments; HG had the highest proportion of AV^+^/PI^+^ and AV^+^/PI^−^ cells; the AV^−^/PI^−^ cells (living cells) showed similar results with the cell viability experiment (Figure 1B).

Then, to explore the dynamics of cell death in the HG, RG, and IG groups, we detected the apoptosis and necrosis rate and the cell proportion in each quadrant of cells from the NG, HG, RG, and IG treatments at the 0th, 6th, 12th, 24th, 36th, and 48th h, respectively. The results are shown in Figure 2. In general, after the 24th h, the apoptosis and necrosis rates (Figure 2A,B) of MDBK cells in the HG and IG groups increased faster than in the NG and RG groups (*p* < 0.05 and *p* < 0.001, respectively), but no difference were found between the HG and IG groups (*p* > 0.05) and between the NG and RG groups (*p* > 0.05). In all quadrants, the IG and HG groups showed a faster increase rate than the NG and RG groups (*p* < 0.05), and no differences were found between the HG and IG groups (*p* > 0.05) and between the NG and RG groups (*p* > 0.05) (Figure 2C–F). Intriguingly, the cell proportion of AV^−^/PI^−^ cells in the NG group was slightly higher than in the RG group (*p* = 0.063).

Until now, we confirmed that the HG and IG stimulation would induce injury to the cultured MDBK cells. To explore the mechanisms involved in the HG and IG-induced MDBK cell injury, we first detected the ROS accumulation in the cultured MDBK cells, and the results are shown in Figure 3A. The relative ROS levels of the HG and IG groups were significantly higher than the NG and RG groups (*p* < 0.05), while ROS levels were similar between HG and IG groups and between NG and RG groups (*p* > 0.05). The relative message RNA (mRNA) expression levels of the nicotinamide adenine dinucleotide phosphate oxidase 4 (*nox4*) gene, which regulates the production of endogenous ROS, are shown in Figure 3B. The *nox4* expression levels of the HG and IG groups were significantly higher than that of the NG group (*p* < 0.05). Then, we pretreated MDBK cells in the HG and IG groups with 1.0 mmol/L (0.1, 0.5, 1.0, and 2.0 mM were tested, and 1.0 mM showed the best ROS inhibitory effect, data did not show) ROS inhibitor NAC for 30 min. The results are shown in Figure 3C,D. Pretreatment with 1.0 mM NAC didn’t affect the ROS level (*p* > 0.05) or *nox4* gene expression level (*p* > 0.05) in the NG group. However, pretreatment with 1.0 mM NAC decreased cellular ROS levels (*p* < 0.01) and *nox4* gene expression levels (*p* < 0.05) in both HG and IG groups.

Then, we further detected the expression levels of *p53*/*caspase-3-*induced apoptosis pathway-related genes (*p53*, *bax*, *caspase-9,* and *caspase-3*) in the HG and IG treatments. The results are shown in Figure 3E–H. The expression levels of genes *p53*, *bax*, *caspase-9*, and *caspase-*3 in the HG treatment were significantly higher (*p* < 0.01) than in the NG treatment. The expression levels of genes *p53* and *caspase-3* in the IG treatment were significantly higher (*p* < 0.05) than in the NG treatments, while the expression levels of *bax* and *caspase-9* genes were similar between the IG and NG treatments (*p* > 0.05). Then, we also detected these genes’ expression levels in each group after 1.0 mM NAC pretreatment, and the results are shown in Figure 3I–L. NAC pretreatment did not alter these genes’ expression levels (*p* > 0.05) in the NG treatment but extremely decreased their expression levels in the HG treatments (*p* < 0.01). In the IG group, NAC significantly reduced the expression levels of *p53* and *caspase-3,* while no significant differences were observed in *bax* and *caspase-*9.

The results of the expression of p53 protein in each group and each NAC pretreated group are shown in Figure 3M. HG and IG treatments significantly (*p* < 0.01) increased the expression levels of the p53 protein. NAC pretreatment significantly decreased the p53 protein expression in the HG and IG treatments (*p* < 0.01). Then, we detected the apoptosis and necrosis rate in NAC-pretreated NG, HG, and IG treatments, and the results are shown in Figure 3N,O. Pretreatment with 1.0 mM NAC significantly decreased the apoptosis rate in the HG and IG treatment groups (*p* < 0.01), and significantly reduced the necrosis rate in the HG treatment (*p* < 0.01) but no effect was noted in IG treatment (*p* > 0.05).

To explore the inflammatory response in each group, we detected the levels of pro-inflammatory cytokines IL-18 and IL-1β at the 24th and 48th h, and the results are shown in Figure 4A,B. At the 24th h, the levels of IL-18 and IL-1β in each group were all similar (*p* > 0.05). At the 48th h, the levels of IL-18 and IL-1β in the NG, HG, and IG treatments were similar (*p* > 0.05), but the IG treatment had significantly higher levels of IL-18 and IL-1β than other treatments (*p* < 0.05). To explore the mechanisms of the increased IL-18 and IL-1β levels in the IG treatment, we first detected the mRNA (Figure 4C) and protein expression levels of NLRP3 protein (Figure 3N and Figure 4D) in each group. The mRNA and protein expression of NLRP3 in the NG and HG treatments were similar (*p* > 0.05), but the IG treatment group had significantly higher mRNA and protein levels of NLRP3 than other treatments (*p* < 0.01). Then, we detected the mRNA expression levels of genes *txnip* and *p2x4* and the concentration of intracellular K^+^ in each group, and the results are shown in Figure 4E–G. IG treatment significantly increased the mRNA expression levels of the *txnip* gene (*p* < 0.05) and significantly decreased the intracellular K^+^ level (*p* < 0.05), but showed an inapparent effect on the mRNA expression of gene *p2x4* (*p* = 0.0786). In addition, we also explored the mRNA expression levels of downstream genes *caspase-1* and *gsdmd*. Figure 4H,I shows that the mRNA expression levels of gene *caspase-1* and *gsdmd* in the IG treatment were significantly higher than in other treatments (*p* < 0.01).

To further verify that IG treatment-induced inflammatory reaction involves the activation of the *nlrp3*/*caspase-1*/*gsdmd* pyroptosis pathway, we pretreated MDBK cells in the NG and IG treatments with siRNA specifically targeting the *nlrp3* gene (*nlrp3*-siRNA) or non-targeting siRNA (NA-siRNA). As shown in Figure 5A, *nlrp3*-siRNA did not affect the relative mRNA expression levels of the *nlrp*3 gene in the NG treatment group but significantly decreased its level in the IG treatment; NA-siRNA did not affect the expression levels of the *nlrp3* gene in the NG either IG treatments. The relative mRNA expression levels of *nox4*, *p2x4*, and *txnip* genes were not affected by *nlrp3*-siRNA and NA-siRNA in the NG or IG treatments (Figure 5B–D). However, *nlrp3*-siRNA significantly decreased the relative mRNA expression levels of *caspase-1* and *gsdmd* genes in the IG treatments to the levels in the NG treatments; NA-siRNA did not affect their expression levels in the NG or IG treatments (Figure 5E,F).

Then, we pretreated the MDBK cells in the NG and IG treatments with 1 mmol/L NAC, 2 μmol/L 5-BDBD, and 10 nmol/L MCC950, respectively (the selected concentrations were tested, data did not show). NAC pretreatment significantly decreased the relative mRNA expression levels of the *nox4* gene (*p* < 0.05) and ROS (*p* < 0.01) in the IG treatment while having no effects on the NG treatment, and 5-BDBD and MCC950 did not affect the mRNA expression levels of *nox4* gene and ROS level in either the NG or IG treatments (Figure 6A,B). 5-BDBD pretreatment significantly decreased the relative mRNA expression levels of the *p2x4* gene (*p* < 0.01) and increased the intracellular K^+^ levels (*p* < 0.01) in the IG treatment, while no effects on the NG treatment and NAC and MCC950 did not affect the expression levels of *p2x4* gene and K^+^ level in the NG and IG treatments (Figure 6C,D). Then, we also detected the mRNA expression levels of *txnip*, *nlrp3*, *caspase-1*, and *gsdmd* gene (Figure 6E–H). NAC-pretreatment significantly inhibited the IG treatment-induced overexpression of all these four genes (*p* < 0.01) to the normal level (against NG treatment, *p* > 0.05); 5-BDBD and MCC950 pretreatment significantly decreased the mRNA expression of *txnip* gene in the IG group (*p* < 0.05) but still higher than in NG group (*p* < 0.05); however, 5-BDBD and MCC950 pretreatment lowered the mRNA expression levels of *nlrp3*, *caspase-1*, and *gsdmd* genes in the IG treatment (*p* < 0.01) to the normal level (against NG treatment, *p* > 0.05). All three inhibitors significantly inhibited the IG treatment-induced overexpression level of NLRP3 protein (*p* < 0.001) to the normal level (against NG treatment, *p* > 0.05, Figure 3N and Figure 6I). We also detected the levels of pro-inflammatory cytokines IL-18 and IL-1β pretreated with these three inhibitors, and these inhibitors significantly decreased the IG treatment-induced high concentration of IL-18 and IL-1β (*p* < 0.05) to normal level (against NG treatment, *p* > 0.05, Figure 6J,K). NAC pretreatment significantly increased the cell viabilities (*p* < 0.05) to the normal level, while 5-BDBD and MCC950 had no effects on the cell viabilities in the IG treatment (Figure 6L). Finally, NAC pretreatment significantly lowered the rate of PI^+^ cells in the IG treatment to the NG treatment (*p* < 0.01); 5-BDBD and MCC950 significantly decreased the rate of PI^+^ cells in the IG treatment (*p* < 0.05) but still higher than in the NG treatment (*p* < 0.05, Figure 6M).

## 3. Discussion

DN is a major microvascular complication of DM and is becoming a heavy global burden on the health career of the world [2]. This microvascular complication develops in approximately 40% and 30% of patients with type 2 DM and type 1 DM, respectively [22]. DN is thought to be emerging widely as the primary factor of end-stage renal disease, and approximately 42% of the patients who suffered end-stage renal disease had a diagnosis of ND in the USA [2,23]. The damage in DN patients’ tissue is irreversible, making prevention of the occurrence and progress of DN the best choice [24]. However, the exact mechanisms of DN are still ambiguous, and it is urgent to understand the underlying mechanisms to develop new strategies to prevent DN.

### 3.1. Abnormal Glucose Affected the Viability, Apoptosis and Necrosis Rate of the Cultured MDBK Cells

DN is suggested to be involved in hemodynamic alterations, including hyperglycemia, hypoglycemia, and glycemia variation [25]. Our study determined abnormal glycemia conditions that occurred in the blood of postpartum dairy cows, including hyperglycemia, hypoglycemia, and glycemia variation, using 2.5 mM, 25 mM, and 2.5~25 mM glucose in the medium, respectively, to explore their effects on the MDBK cells, a kind of renal epithelial cells isolated from Holstein dairy cows. The previous study showed that 5 and 25 mM mannitol did not affect the viability of Tohoku Hospital Pediatrics-1 (THP-1) cells [26], indicating that the difference in cell viability between the NG, HG, and IG groups was induced by glucose concentration difference rather than by osmotic pressure difference. In our study, the viabilities in the LG and RG group were similar to the NG treatment (*p* > 0.05, Figure 1B), indicating that 48 h of hypoglycemia and the process of glycemia decrease had no significant detrimental effects on the MDBK cells. However, HG and IG treatments significantly decreased the viabilities (*p* < 0.05, Figure 1B), and the annexin V-FITC/PI apoptosis assay further showed that HG and IG treatments significantly increased the apoptosis rate and necrosis rate of MDBK cells (*p* < 0.05, Figure 1C–E), indicating they might be the factors which cause damage and inflammatory response to the renal glomerulus epithelium. Interestingly, the cell viability in the HG group was lower than in the IG group, and the apoptosis rate in the IG group was lower than in the HG group (Figure 1B–D), indicating that HG and IG treatments caused damage to the MDBK cells by different mechanisms. Li et al. found that 30 mM glucose significantly increased the apoptosis rate of mouse podocyte clone 5 cells via the Nox4/ROS pathway [27]. Intermittent high glucose increased pro-inflammatory cytokines IL-1β and IL-18 in the monocytic cells [26,28] and human umbilical endothelial cells [29]. Together with our results, we assume that the mechanisms of HG and IG treatments induced MDBK cell injury might involve *nox4*/ROS-mediated apoptosis and *nlrp3*/*caspase-1*/*gsdmd*-mediated pyroptosis pathways, respectively.

### 3.2. The nox4/ROS/p53/caspase-3 Mediated Apoptosis Pathway Participated in the MDBK Cells Injury Induced by Abnormal Glucose

We assumed that HG caused damage to MDBK cells via the apoptosis pathway. We explored the apoptosis rate at different time points. From the 24th h, the apoptosis and necrosis rates in the HG and IG groups were significantly higher than in the NG and RG groups (Figure 2A,B), indicating that constant hyperglycemia enhances the apoptosis pathway in the renal epithelial cells. The relative mRNA expression levels of the *p53*/*caspase-3* apoptosis pathway (*p53*, *bax*, *caspase-9*, and *caspase-3*) were significantly higher in the HG group than NG group (Figure 3E–H), and the p53 protein expression level was also considerably higher in the HG group than NG group (Figure 3N), supporting the key role of chronic hyperglycemia in the renal glomerulus dysfunction [15,30,31]. The *nox4*/ROS is an important stimulation pathway of the *p53*/*caspase-3*-mediated apoptosis pathway [27,32]. In the present study, HG treatment significantly increased the relative mRNA expression level of the *nox4* gene and the level of cellular ROS (Figure 3A,B). The ROS scavenger NAC blocked HG treatment-induced activation of the *nox4*/ROS/*p53*/*caspase-3*-mediated apoptosis pathway and decreased the apoptosis and necrosis rate of MDBK cells (Figure 3C–O). These results indicated that the *nox4*/ROS/*p53*/*caspase-3*-mediated apoptosis pathway was a major contributor to the chronic hyperglycemia-induced renal injury. It must be noticed that, in Figure 1B, the cell viability in the NG was higher than in the RG group (*p* < 0.05); however, they were similar in Figure 2F (*p* = 0.063), which might be explained by random error.

It is worth noting that HG treatment extremely increased the proportion of AV^−^/PI^+^ cells from the 24th h (Figure 3C), and AV^−^/PI^+^/high caspase-1 was thought to be a biomarker of pyroptosis [33], indicating that the cell membrane integrity might be damaged via pyroptosis pathway [34]. In the present study, the relative mRNA expression level of *caspase-1* in the HG treatment was slightly increased (*p* < 0.07, Figure 4H), supporting the activation of the pyroptosis pathway in the HG treatment. Similarly, the classical pyroptosis pathway-related gene (*p2x4*, *txnip*, *nlrp3*, and *gsdmd*), protein (NLRP3) levels, and the IL-1β and IL-18 levels in the HG treatment were slightly higher than in NG treatment, and the cellular K^+^ level in the HG group was marginally lower than NG treatment (Figure 4). These results showed that the pyroptosis pathway was not fully activated at the 48th h in the HG treatment. However, from the 24th to the 48th, the proportion of AV^−^/PI^+^ cells in the HG treatment was significantly higher than in the NG treatment, which is inconsistent with the results in Figure 4, and this contradiction needs to be further explored. Anyway, it is clear that the *nox4*/ROS/*p53*/*caspase-3*-mediated apoptosis pathway is involved in the HG treatment-induced renal glomerulus epithelial cells injury. The possible mechanisms of HG treatment-induced MDBK cell injury are illustrated in Figure 7.

### 3.3. Glucose Fluctuation Induced ROS/nlrp3/caspase-1/gsdmd Mediated Pyroptosis Pathway in MDBK Cells

A previous study showed intermittent high glucose-induced pyroptosis in the cardiomyocyte [35], and the role of pyroptosis in the DN was reviewed recently [14,36]. In the present study, IG treatment significantly increased the levels of pro-inflammatory cytokines IL-1β and IL-18, the relative mRNA expression levels of clinical pyroptosis pathway-related genes (*p2x4*, *txnip*, *nlrp3*, *caspase-1*, and *gsdmd*), and NLRP3 protein level (Figure 4). Knockdown of the *nlrp3* gene and inhibition of NLRP3 protein significantly reduced the relative mRNA expression level of downstream *gsdmd* gene, the levels of pro-inflammatory cytokines IL-1β and IL-18, and the proportion of PI^+^ cells, and increased the viabilities in the IG group (Figure 5 and Figure 6). These results confirmed the role of the pyroptosis pathway in the IG treatment-induced MDBK cells injury, supporting the opinion that glycemia variation plays an important role in the DN [17,18] via the *nlrp3*/*caspase-1*/*gsdmd*-mediated pyroptosis pathway.

Activating the NLRP3 inflammasome, whose activation can be stimulated by a series of endogenous and exogenous factors, is pivotal for the clinical pyroptosis pathway [37]. ROS derived from mitochondria and cellular K^+^ efflux are the mediators of NLRP3-mediated inflammasome complex activation during the development of the DN [38]. In the present study, NAC and 5-BDBD treatments, scavenger of ROS and inhibitor of P2X4, which are the ion channel proteins that transport K^+^ outward [39], respectively, decreased the relative mRNA levels of the *caspase-1* and *gsdmd* gene (Figure 6G,H). In addition, MCC950 treatment, the inhibitor of NLRP3 protein, showed a stronger inhibitory effect on the expression of the *gsdmd* gene (Figure 6H), indicating that the ROS and cellular K^+^ efflux both participated in the activating of NLRP3 inflammasome. It must be noted that the formation of the GSDMD-N protein pore might also non-selectively enhance the efflux of cellular K^+^, which increases the activation of the NLRP3 inflammasome [40]. However, in our study, 5-BDBD treatment increased the intracellular K^+^ level in the IG group to the level in the NG group (Figure 6D), showing that inhibition of P2X4 protein could block the process of intracellular K^+^ efflux, indicating that the GSDMD-N protein pore did not involve in the process of intracellular K^+^ efflux. However, this conclusion needs to be further investigated. Another noteworthy point is that NAC treatment restored the viabilities and proportion of PI^+^ cells in the IG group, while 5-BDBD and MCC950 treatments did not block the effect of IG treatment (Figure 6L,M). These results indicate that ROS/*txnip*/*nlrp3*/*caspase-1*/*gsdmd*-mediated pyroptosis plays a dominant role in IG treatment-induced MDBK cell injury. The possible mechanisms of IG treatment-induced MDBK cell injury are illustrated in Figure 7.

In the present study, LG treatment showed non-significant deleterious effects on the MDBK cells (Figure 1B). However, Luo et al. found that recurrent hypoglycemia impaired vascular function in advanced diabetes mellitus rats by inducing pyroptosis [41] and hypoglycemia was also found to increase the risk of motility in patients with kidney disease [42]. We assumed that these results could also be explained by the fact that the shift in the blood glucose contained the IG process. It is a nonnegligible limitation that the cells we used are immortalized renal glomerulus epithelial cells instead of primary cells, and whether the same phenotype occurs in the isolated primary cells or in vivo must be verified. Furthermore, the treatment employed in our study referred to previous studies, which could not represent the complex situations in the blood of diabetic patients. More in vivo studies need to be carried out to verify our observation.

## 4. Materials and Methods

The MDBK cell line was purchased from China Center for Type Culture Collection (CCTCC). All the agents used in this experiment were purchased from Solarbio Technology Co., Ltd., Beijing, China, unless there is a specific statement.

### 4.1. Experiment Design and Cell Culture

The MDBK cells were maintained in Dulbecco’s modified eagle medium (DMEM) containing 5% fetal bovine serum (FBS) (ThermoFisher, Shanghai, China), 100 U/mL penicillin, 100 μg/mL streptomycin. Immortalized bovine renal glomerulus epithelial (MDBK) cells were incubated at 37 ℃ and 5% CO_2_. Referencing a previous study [26], we stimulated the MDBK cell with five treatments to mimic abnormal glycemia, including low glucose (LG), 2.5 mM glucose for 48 h; normal glucose (NG), 5 mM glucose for 48 h; high glucose (HG), 25 mM glucose for 48 h; reducing glucose (RG), 25 mM glucose for 24 h followed by 2.5 mM glucose for 24 h; increasing glucose (IG), 2.5 mM glucose for 24 hg followed by 25 mM glucose for 24 h (Figure 1A). In the experiments involving inhibitors, the MDBK cells (60~70% confluent) were pretreated for 30 min with each inhibitor dissolved in 100% dimethyl sulfoxide (DMSO; Sigma-Aldrich) and then diluted in a culture medium. N-acetyl-L-cysteine (NAC) was purchased from Solarbio Technology Co., Ltd., Beijing, China, and C17H11BrN2O2 (P2X receptor family 4 (P2X4) receptor antagonist, 5-BDBD) and MCC950 sodium salt (NOD-like receptor thermal protein domain-associated protein 3 (NLRP3) inflammasome inhibitor, MCC950) were purchased from Sigma Chemical Co., St. Louis, MO, USA.

### 4.2. Cell Viability Assay

MDBK cells of each treatment were plated into 96-well flat-bottom plates (Nest, China) at a density of 10^4^ cells/well and incubated at 37 °C with 5% CO_2_ until the cells grew to approximately 60% confluence. Then, the MDBK cultured cells’ viabilities were detected using cell counting kit-8 (Nanjing Jiancheng, Nanjing, China) according to the manufacturer’s instructions.

### 4.3. Real-Time Quantitative Polymerase Chain Reaction

The total ribonucleic acid (RNA) of cultured MDBK cells was extracted from each treatment using Trizol reagent (ThermoFisher, Shanghai, China). The extracted total RNA was reverse-transcribed into complementary deoxyribonucleic acid (cDNA) using a reverse transcription kit (Takara, Tokyo, Japan). The real-time quantitative polymerase chain reaction (RT-qPCR) process was performed in a 10 μL reaction system per well in a 96-well plate containing 0.08 μL diluted cDNA, 5 μL SYBR^®^ Premix Ex Taq II (Tli RNASEH Plus), 0.04 μL ROX Reference Dye, 0.3 μL each upstream and downstream primers (10 μM), and 3.56 μL ddH2O. The RT-qPCR amplification procedure was as follows: 95 °C 30 s; 95 °C 5 s, 61 °C 30 s, 39 cycles; melt curve: 65 °C → 95 °C. The RT-qPCR reactions were performed on the CFX96 Touch Real-Time PCR Detection System (Bio-Rad, CA, USA) using a Real-Time PCR Kit (Takara, Tokyo, Japan). The sequences of used primers are shown in Appendix A. The data generated by RT-qPCR were assessed via the 2^−ΔΔCt^ method, and the relative gene expression level was calculated using β-actin for normalization.

### 4.4. Immunoblot Assay

The cultured MDBK cells of each treatment were collected and lysed in a lysis buffer containing proteinase inhibitors, phosphatase inhibitors, and phenylmethanesulfonyl fluoride. Later, the lysates were centrifuged, and the supernatants were quantified via the BCA Protein Assay Kit (Tiangen, Beijing, China). The same amounts of protein were placed into each lane of sodium dodecyl sulfate-polyacrylamide gels and separated by electrophoresis. The separated proteins were then transferred to polyvinylidene fluoride membranes through an electron-transfer method using the Bio-Rad Trans-Blot apparatus (Bio-Rad, Hercules, CA, USA). The polyvinylidene fluoride (PVDF) membranes were immersed in 5% skim milk for subsequent blocking, incubated with primary antibodies, hybridized with matched secondary antibodies, and then incubated with enhanced chemiluminescent reagents to visualize the bands of proteins. The band intensity of the images was measured using the Tanon-5200 automatic chemiluminescence imaging analysis system (Tanon, Shanghai, China).

### 4.5. Annexin V-FITC/PI Apoptosis Assay

The apoptosis and necrosis rate of the MDBK cells in each treatment referenced a previous study [43], and the flow cytometry was used to perform the annexin V-FITC/PI apoptosis assay using the Annexin V-FITC/PI Apoptosis Kit (Nanjing Jiancheng, Nanjing, China). Briefly, the cultured MDBK cells were dissociated by trypsin digestion and washed with ice-cold PBS. The washed MDBK cells were collected, resuspended in binding buffer, and added with annexin V-FITC/PI solution. After 15 min of cultivation at room temperature in the dark, the cells were assessed through a FACScan flow cytometry system. Cells in the AV^+^/PI^−^ and AV^+^/PI^+^ quadrants were considered under an apoptotic state and necrosis state, respectively [43]. Cells in the AV^−^/PI^+^ with high levels of *caspase-1* expression were considered under a pyroptosis state [33], and cells in the AV-/PI- quadrant were considered health cells.

### 4.6. Enzyme-Linked Immunosorbent Assay

The levels of ROS and K^+^ in the MDBK cells and pro-inflammatory cytokines, including interleukin (IL)-18 and IL-1β, in the supernatants of cultured MDBK cells were detected using enzyme-linked immunosorbent assay (ELISA) kits (Nanjing Jiancheng, Nanjing, China) according to the manufacturer’s instructions.

### 4.7. Cell Transfection

The siRNA-specific targeting and non-targeting *nlrp3* were synthesized by Hanbio Technology Co., Ltd., Shanghai, China. The sequences of the *nlrp3*-target siRNA were 5′-AGAUGCAUUUAGAAGACUA-3′ (sense) and 5′-UAGUCUUCUAAAUGCAUCU-3′ (antisense), the sequences of the nontarget siRNA were 5′-UUCUCCGAACGUGUCACGU-3′ (sense) and 5′-ACGUGACACGUUCGGAGAA-3′ (antisense). Referencing a previous siRNA transfection study in MDBK cells [44], we transfected the siRNAs into the MDBK cells using Opti-MEM medium (ThermoFisher, Shanghai, China) and TransIntro EL Transfection Reagent (TransGen, Beijing, China) according to the manufacturer’s instructions. The downregulation of the *nlrp3* gene was confirmed by RT-qPCR after 48 h of transfection.

### 4.8. Statistical Analysis

The results of all experiments were expressed as the mean ± standard deviation. Statistical analysis was implemented using SPSS 26 (IBM, New York, NY, USA), and graphs were prepared in GraphPad Prism 9.0 (GraphPad Software, CA, USA). Student *t-*test was used for two-group comparisons, and one-way ANOVA was used when three or more groups were compared. General linear model-repeated measures (GLM) were used to evaluate the effects of culture time and glucose concentration on MDBK cells. Differences were considered statistically significant when *p* < 0.05 and extremely significant when *p* < 0.01.

## 5. Conclusions

In conclusion, the low glucose and process of glucose decreases showed non-significant effects on the viability, apoptosis, and necrosis rate of the MDBK cells. However, 24h of high glucose increased the level of ROS and activated the *p53*/*caspase-3*-mediated apoptosis pathway. The process of glucose increases upregulated the expression levels of *txnip*/*nlrp3*/*caspase-1*/*gsdmd*-mediated pyroptosis pathway-related genes, which were evoked by ROS and K^+^ efflux. Our results indicated that hyperglycemia and glycemia fluctuation, especially the process of glycemia increases, might cause injury to the renal glomerulus epithelial cells and lead to nephritis. Our study showed that the process of glycemia increases was mainly responsible for the nephritis in diabetic nephropathy, underlining the importance of glycemic control in diabetes patients.

## Figures and Tables

**Figure 1 ijms-24-00600-f001:**
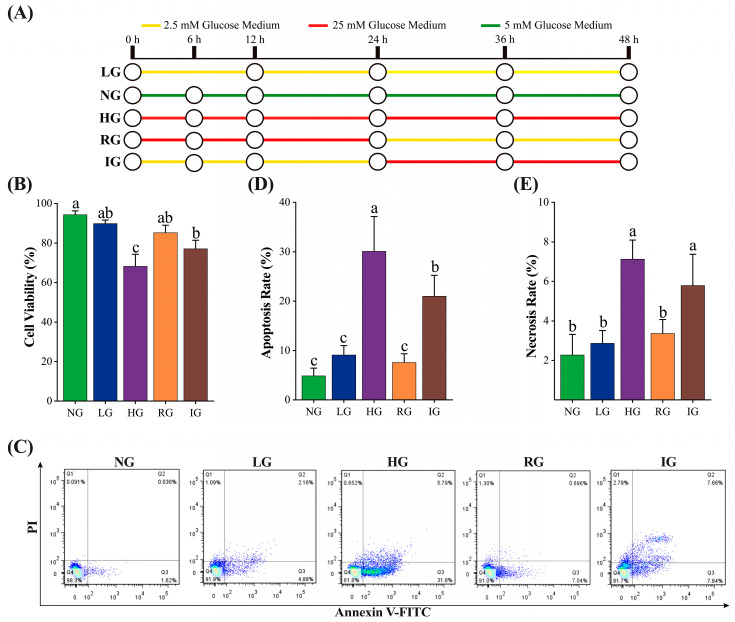
The effects of abnormal glucose on the viability of cultured MDBK cells. (**A**) Schematic diagram of experimental design; (**B**,**D**,**E**) the effects of abnormal glucose on the cell viability, apoptosis rate, and necrosis rate of the cultured MDBK cells, respectively. (**C**) The flow cytometry results of the annexin V-FITC/PI assay. All experiments were repeated at least 3 times, and the data were presented using mean ± SD. Histograms with different superscript letters (a–c) are significantly different (Waller–Duncan, *p* < 0.05).

**Figure 2 ijms-24-00600-f002:**
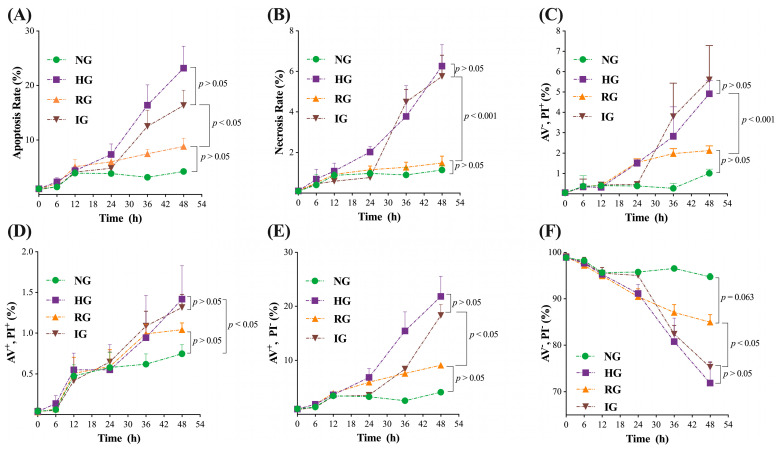
The dynamics of cell death in the HG and IG groups. (**A**,**B**) The apoptosis and necrosis rate of cultured MDBK cells at the 0th, 6th, 12th, 24th, 36th, and 48th h. (**C**–**F**) The cell proportion in each quadrant at the 0th, 6th, 12th, 24th, 36th, and 48th h. All experiments were repeated 3 times, and the data were presented using mean ± SD. General linear model-repeated measures.

**Figure 3 ijms-24-00600-f003:**
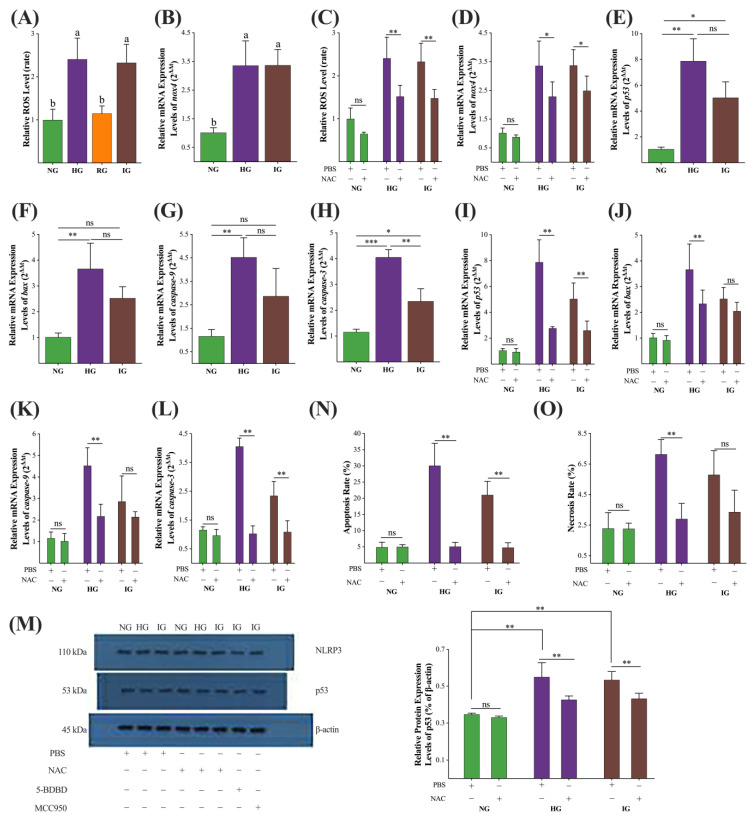
The *nox4*/ROS evoked *p53*/*caspase-3* apoptosis pathway participates in the abnormal glucose-induced MDBK cell injury. (**A**) the cellular ROS levels in the NG, HG, RG, and IG groups; (**B**–**D**) the *nox4* mRNA expression levels in each group and their NAC pretreated corresponding groups; (**E**–**L**) the relative expression levels of *p53*, *bax*, *caspase-9*, and *caspase-3* genes in NG, HG, and IG groups and their NAC pretreated corresponding groups; (**M**), the gel picture and grayscale comparison of immunoblot assay and the relative expression levels of p53 protein in the NG, HG, and IG groups and their NAC pretreated affiliated groups; (**N**,**O**) the apoptosis and necrosis rates in the NG, HG, and IG groups and their NAC pretreated corresponding groups. All experiments were repeated 3 times, and the data were presented using mean ± SD. Histograms with different superscript letters (a–b) are significantly different (Waller–Duncan, *p* < 0.05). *, *p* < 0.05; **, *p* < 0.01; ***, *p* < 0.001; ns, non-significant.

**Figure 4 ijms-24-00600-f004:**
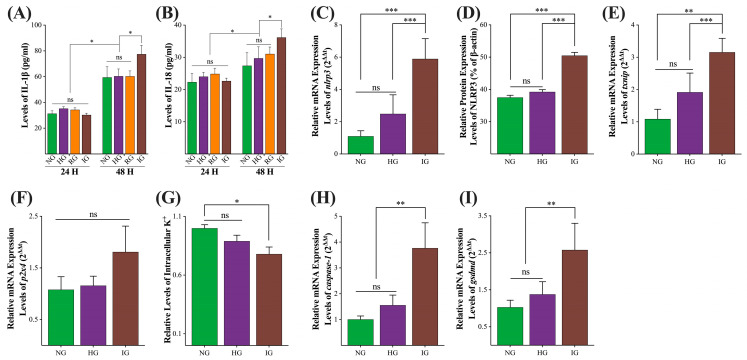
The effects of abnormal glucose on the *nlrp3*/*caspase-1*/*gsdmd* pyroptosis pathway. (**A**,**B**) The levels of pro-inflammatory cytokines IL-18 and IL-1β in the supernatant of each group; (**C**,**E**–**I**) the relative mRNA expression levels of *nlrp3*, *txnip*, *p2x4*, *caspase-1*, and *gsdmd* genes in each group; (**D**) the relative protein levels of NLRP3 protein in each group; (**G**) the levels of intracellular K^+^ in each group. All experiments were repeated 3 times, and the data were presented using mean ± SD. Two-tailed *t*-test; *, *p* < 0.05; **, *p* < 0.01; ***, *p* < 0.001; ns, non-significant.

**Figure 5 ijms-24-00600-f005:**
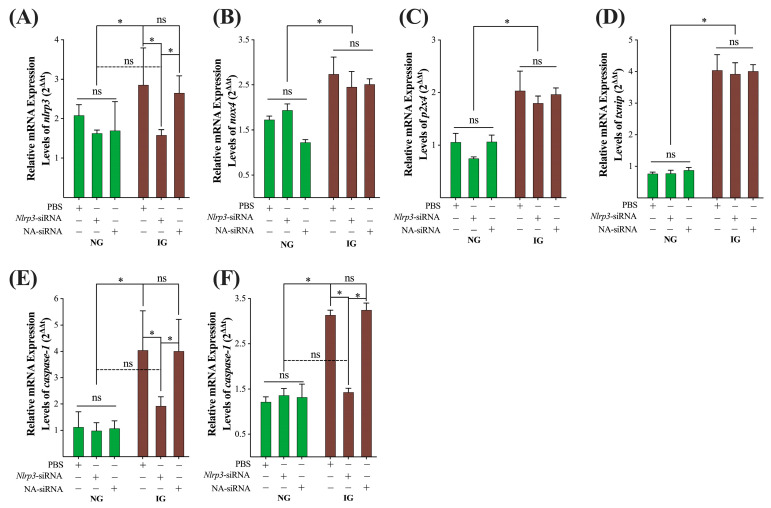
Knockdown of the *nlrp3* gene inhibits the IG treatment-induced activation of the *nlrp3*/*caspase-1*/*gsdmd* pyroptosis pathway. (**A**–**F**) the relative mRNA expression levels of MDBK cells treated with siRNA targeting the *nlrp3* gene and control siRNA. All experiments were repeated 3 times, and the data were presented using mean ± SD. Two-tailed *t*-test; * *p* < 0.05; ns, non-significant.

**Figure 6 ijms-24-00600-f006:**
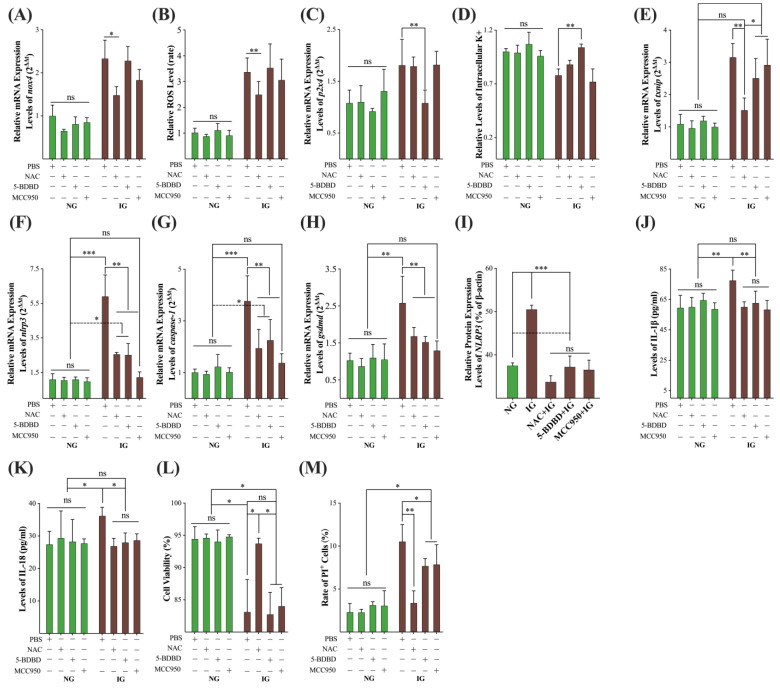
Inhibitors blocked the IG treatment-induced activation of the *nlrp3*/*caspase-1*/*gsdmd* pyroptosis pathway. (**A**,**B**) The effects of inhibitors on the mRNA expression levels of the *nox4* gene and intracellular ROS levels; (**C**,**D**) the effects of inhibitors on the mRNA expression levels of the *p2x4* gene and intracellular K^+^ levels; (**E**–**H**) the impact of inhibitors on the mRNA expression levels of *txnip*, *nlrp3*, *caspase-1*, and *gsdmd* genes; (**I)** the effects of inhibitors on the expression levels of the NLRP3 protein; (**J**–**M**) the impact of inhibitors on the IL-1β, IL-18, cell viability, and rate of PI^+^ cells, respectively. All experiments were repeated 3 times, and the data were presented using mean ± SD. ANVOA, *, *p* < 0.05; **, *p* < 0.01; ***, *p* < 0.001; ns, non-significant.

**Figure 7 ijms-24-00600-f007:**
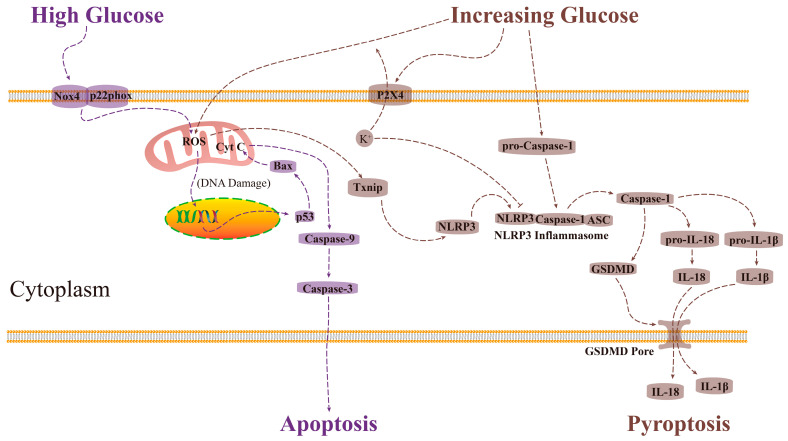
The sketch map of abnormal glucose-induced MDBK cell injury. High glucose increased the level of Nox4 protein in the cell membrane, which elevated the ROS level in the mitochondria. The elevated ROS damaged the DNA, activating the p53 protein. The p53 protein induced the assembly of Bax protein in the mitochondrial membrane, releasing the Cyt C protein, which triggered the downstream proteins caspase-9 and caspase-3, resulting in apoptosis. Increasing glucose elevated the ROS level in the mitochondria, which activated the Txnip protein and then activated the NLRP3 protein. Increased glucose also increased the level of P2X4 protein in the cell membrane, resulting in the decrease of the concentration of intracellular K^+^, which inhibited the activation of the NLRP3 inflammasome. Increasing high glucose also increased the level of pro-caspase-1, which participated in the assembly of the NLRP3 inflammasome. The released caspase-1 protein from the NLRP3 inflammasome promoted the self-splicing of GSDMD protein, pro-IL-I8, and pro-IL-1β. The GSDMD-N formatted a non-selective protein pore, which allowed the efflux of the mature IL-18 and IL-1β, resulting in cell pyroptosis and aggressive inflammatory response.

## Data Availability

The data used to support the findings of this study are available from the corresponding author upon request.

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
