# Peer review of "Process of Glucose Increases Rather Than Constant High Glucose Was the Main Cause of Abnormal Glucose Induced Glomerulus Epithelial Cells Inflammatory Response"

_ijms, 2022, doi:10.3390/ijms24010600_

Round 1

Reviewer 1 Report

Interesting and nice research, congrats to the authors. However, there are some minor issues that need to be addressed before it can be published, and the specific comments are:

1. careful English checking is needed to correct grammar and typo errors. As in line 92: Please change ‘24 hg’ to be ‘24h’.

2.      Author must add references for every method.

3.      Figure  3N does not justify the claim in the result section. Authors must replace figure 3N with a good quality image.

4.      Authors must check the reference error throughout the manuscript. 

Author Response

Thank you very much for your comments and advice, and your advice increased the quality of our manuscript very much. According to your suggestion, we’ve revised our manuscript carefully, and the responses are as follows:

Comment 1: careful English checking is needed to correct grammar and typo errors. As in line 92: Please change ‘24 hg’ to ‘24h’.

Response 1: I am so sorry for this kind of mistake in the manuscript. In the revised manuscript, we have corrected this mistake (line 159) and carefully checked all the text throughout the manuscript.

Comment 2: The author must add references for every method.

Response 2: Following your advice, we added references for the flow cytometry analysis and siRNA transfection experiment (lines 232, 239, and 240). If you think other methods (cell viability detection, RT-qPCR, WB, ELISA) also need support, we will add corresponding references in the next revision.

Comment 3: Figure 3N does not justify the claim in the result section. Authors must replace figure 3N with a good-quality image.

Response 3: I am sorry for the low-quality gel picture of our WB experiment. Maybe it was because we didn't use a high-resolution camera when we took pictures. According to your advice, we have replaced the gel picture with a higher-definition gel picture (line 536). If it’s necessary, we could upload the analysis picture in the next revision.

Comment 4: Authors must check the reference error throughout the manuscript.

Response 4: I am sorry for these garbled codes (Error! Reference source not found) throughout the manuscript. These garbled codes should have been the cross-references of the Figures, and I am unaware of why these codes occurred. In the revised manuscript, we did not use the cross-references function, and these garbled codes have been removed (lines 159, 279, 284, and 287, et al.)

Reviewer 2 Report

In this manuscript, authors reported that increasing glucose concentration rather than constant high glucose concentration induced the production of proinflammatory cytokines including IL-1beta and IL-18 in immortalized bovine renal glomerulus epithelial (MDBK) cells, possibly through activating txnip/nlrp3/gsdmd mediated pyroptosis pathway. The subject of study seems to be interesting. However, there are several concerns in this study. The reviewer’s comments are described as follows.

1. The title appears to be obviously exaggerated. This study never evaluated hyperglycemia-induced nephropathy but only showed high glucose induced cell injury. This title cannot be accepted.

2. Why authors used MDBK cells in this study remained unclear. If they would like to evaluate glucose-induced podocyte injury, they should use completely differentiated podocytes. Authors have to use an appropriate cell type as a specific target in diabetic nephropathy.

3. The effects of RG and IG on apoptosis, ROS production, and proinflammatory cytokine production have to be evaluated on both 24 and 48 hours in order to show the greater effects of glucose concentration in the latter half period.

4. In the method section, authors did not describe how they defined necrosis and pyroptosis in flow cytometric analysis.

5. Since Figure 4 showed the greater effects of glucose fluctuation on proinflammatory cytokine production, the effects of not only IG but also RG should be examined.

6. There are no error bars in Fig 2.

7. Caspase 3 and 9 should be evaluated on western blot as protein levels rather than mRNA levels in conjugation with cleaved caspase 3 and 9 expressions

8. In the text, there are many inappropriate sentences “Error! Reference source not found”. This error must be corrected.

Author Response

Thank you very much for your interest and comments, and your advice significantly increased the quality of our manuscript. According to your advice, we’ve revised our manuscript carefully, and the responses are as follows:

Comment 1: The title appears to be obviously exaggerated. This study never evaluated hyperglycemia-induced nephropathy but only showed high glucose induced cell injury. This title cannot be accepted.

Response 1: According to your advice, we’ve carefully changed the title of our manuscript to “Process of Glucose Increases rather than Constant High Glucose Was the Main Cause of Abnormal Glucose Induced Glomerulus Epithelial Cells Inflammatory Response” (lines 2-4).

Comment 2: Why authors used MDBK cells in this study remained unclear. If they would like to evaluate glucose-induced podocyte injury, they should use completely differentiated podocytes. Authors have to use an appropriate cell type as a specific target in diabetic nephropathy.

Response 2: The authors are engaged in cattle disease research and noticed that postpartum cows, which frequently suffered glycemic fluctuation, showed a higher frequency of symptoms of nephritis. The authors wondered whether the nephritis in cows was related to glycemic instability. However, no bovine-derived podocyte cell line is available, which is why we used MDBK cells in this study. Surprisingly, the process of glucose increases rather than the constantly high glucose was responsible for the inflammatory response. We consulted the diabetic nephropathy-related references, and no similar literature was found. So, we wrote this manuscript to share our interesting results, and we hope this finding will provide a new direction for the study of the pathogenesis of diabetic nephropathy. We will try to isolate completely differentiated podocytes from cattle in our further studies to deeply explore the specific mechanisms and use a podocyte cell line of mice or human beings to testify the mechanism.

Comment 3: The effects of RG and IG on apoptosis, ROS production, and proinflammatory cytokine production have to be evaluated on both 24 and 48 hours in order to show the greater effects of glucose concentration in the latter half period.

Response 3: As you suggested, we characterized the dynamics of MDBK cells in the NG, HG, RG, and IG groups at the 0th, 6th, 12th, 24th, 36th, and 48th h, respectively, and similar results were obtained and shown in Figure 2 (line 445). These results were described in the RESULTS section (lines 291-299) and discussed in the DISCUSSION section (lines 986-1073). The experimental design schematic diagram was also renewed in Figure 1 A (line 172). The ROS, IL-1beta, and IL18 levels in the 48th h were also detected as per your suggestion, and the results were shown in Figure 3 A (line 536), Figure 4 A and B (line 752), respectively.

Comment 4: In the method section, the authors did not describe how they defined necrosis and pyroptosis in flow cytometric analysis.

Response 4: As per your suggestion, we described how we defined necrosis and pyroptosis in the MATERIALS AND METHODS section (lines 238-241) and added the corresponding references in the revised manuscript (lines 239 and 240).

Comment 5: Since Figure 4 showed the greater effects of glucose fluctuation on proinflammatory cytokine production, the effects of not only IG but also RG should be examined.

Response 5: According to your suggestion in comment 3 and this comment, we detected the IL-18 and IL-1beta levels in the NG, HG, RG, and IG groups at the 24th h and 48th h, and the results were shown in Figure 4A and B in the revised manuscript (line 752).

Comment 6: There are no error bars in Fig 2.

Response 6: I am sorry for this mistake. I’ve added the error bars in Figure 2 in the revised manuscript (line 445).

Comment 7: Caspase 3 and 9 should be evaluated on western blot as protein levels rather than mRNA levels in conjugation with cleaved caspase 3 and 9 expressions.

Response 7: I am sorry for this experimental design defect. In our original experimental design, we intended to verify the expression levels of Caspase 1, 3, and 9 and their corresponding cleaved proteins. However, we couldn’t purchase their specific antibodies for bovine. Considering that we will further explore the mechanisms underlining the interesting finding, we decided to verify the expression levels of these proteins at the protein levels in podocyte cells of human beings or mice.

Comment 8: In the text, there is many inappropriate sentences “Error! Reference source not found”. This error must be corrected.

Response 8: I am sorry for these garbled codes (Error! Reference source not found) throughout the manuscript. These garbled codes should have been the cross-references of the Figures, and I am unaware of why these codes occurred. In the revised manuscript, we did not use the cross-references function, and these garbled codes have been removed (lines 159, 279, 284, and 287, et al.)

Reviewer 3 Report

Work by Qi et al. is an interesting piece of literature, but it requires some significant changes:

- please use superscripts next to the names referring to the affiliation of individual authors;

- I would like to ask you to change citations throughout the text not to superscript, but to normal font;

- please make sure that all abbreviations used in the publication are explained;

- please explain what does line 92 mean: (Error! Ref- erence source not found); on line 170 Error! Reference source not found. B; same on line 177, 182, 186, 190, 192, 193, 209, 214, 232, 238, 244, 248, 253, 257, 258, 259, 264, 268, 281, 286, 289, 302, 306, 308, 316, 320, 322, 326, 358, 360, 363, 366, 379, 383, 385, 389, 392, 396, 400, 404, 408, 411, 433, 437 etc.

- please explain why figure 1 is included in the materials and methods and is not quoted in the text;

-please change (P<0.01) to a lowercase p;

- please improve the quality of all figures in the text: remove the blue background and check the scale, because in some cases the values on the scale are truncated.

Author Response

Thank you very much for your comments and your advice, which increased the quality of our manuscript very much. I’ve revised our manuscript carefully according to your advice, and the responses are as follows:

Comment 1: please use superscripts next to the names referring to the affiliation of individual authors.

Response 1: According to your advice, the numbers next to the authors’ names in the authors' list have been changed to superscripts (lines 5 and 6).

Comment 2: I would like to ask you to change citations throughout the text not to superscript, but to normal font.

Response 2: According to your advice, all citations in the manuscript have been changed to the normal font (lines 33, 36, 38, and 39, et al.)

Comment 3: please make sure that all abbreviations used in the publication are explained.

Response 3: According to your advice, all the unexplained abbreviations in the manuscript have been explained (lines 153, 169-171, 202, and 204, et al.)

Comment 4: please explain what does line 92 mean: (Error! Reference source not found); on line 170 Error! Reference source not found. B; same on line 177, 182, 186, 190, 192, 193, 209, 214, 232, 238, 244, 248, 253, 257, 258, 259, 264, 268, 281, 286, 289, 302, 306, 308, 316, 320, 322, 326, 358, 360, 363, 366, 379, 383, 385, 389, 392, 396, 400, 404, 408, 411, 433, 437 etc.

Response 4: I am so sorry for the trouble caused by these garbled codes. These garbled codes should have been the cross-references of the Figures, and I am unaware of why these codes occurred. In the revised manuscript, we did not use the cross-references function, and these garbled codes have been removed (lines 159, 279, 284, and 287, et al.)

Comment 5: please explain why figure 1 is included in the materials and methods and is not quoted in the text.

Response 5: I am sorry to confuse you. Actually, Figure 1 was quoted many times in the manuscript. It was displayed as “Error! Reference source not found” in the text. In the revised manuscript, this error has been corrected (lines 159, 279, 284, and 287, et al.)

Comment 6: please change (P<0.01) to a lowercase p.

Response 6: According to your advice, all the statistically significant -related “P” has been changed to a lowercase “p” (lines 20, 21, 23, and 24, etc.)

Comment 7: please improve the quality of all figures in the text: remove the blue background and check the scale, because in some cases the values on the scale are truncated.

Response 7: According to your advice, the blue background of all figures has been removed, and the scale of all figures has been adjusted to avoid truncation (lines 178, 449, 536, 748, 771, 914, and 1088).

Round 2

Reviewer 2 Report

Authors have successfully addressed most of the reviewer's concerns in the revised manuscript. The authors' response to the comment #2 should be reflected in the text.

Reviewer 3 Report

Thank you very much for all the changes you made. Now the article is much more readable.